# Visual Detection of Water Content Range of Seabuckthorn Fruit Based on Transfer Deep Learning

**DOI:** 10.3390/foods12030550

**Published:** 2023-01-26

**Authors:** Yu Xu, Jinmei Kou, Qian Zhang, Shudan Tan, Lichun Zhu, Zhihua Geng, Xuhai Yang

**Affiliations:** 1College of Mechanical and Electrical Engineering, Shihezi University, Shihezi 832000, China; 2Engineering Research Center for Production Mechanization of Oasis Characteristic Cash Crop, Ministry of Education, Shihezi 832000, China; 3Xinjiang Production and Construction Corps Key Laboratory of Modern Agricultural Machinery, Shihezi 832000, China

**Keywords:** transfer deep learning, sea buckthorn, moisture content, vision inspection

## Abstract

To realize the classification of sea buckthorn fruits with different water content ranges, a convolution neural network (CNN) detection model of sea buckthorn fruit water content ranges was constructed. In total, 900 images of seabuckthorn fruits with different water contents were collected from 720 seabuckthorn fruits. Eight classic network models based on deep learning were used as feature extraction for transfer learning. A total of 180 images were randomly selected from the images of various water content ranges for testing. Finally, the identification accuracy of the network model for the water content range of seabuckthorn fruit was 98.69%, and the accuracy on the test set was 99.4%. The program in this study can quickly identify the moisture content range of seabuckthorn fruit by collecting images of the appearance and morphology changes during the drying process of seabuckthorn fruit. The model has a good detection effect for seabuckthorn fruits with different moisture content ranges with slight changes in characteristics. The migration deep learning can also be used to detect the moisture content range of other agricultural products, providing technical support for the rapid nondestructive testing of moisture contents of agricultural products.

## 1. Introduction

Seabuckthorn fruit is the dried and mature fruit of seabuckthorn, a plant of the Elaeagnaceae family. It is a perennial deciduous shrub or small arbor. It is a small berry plant that likes light, is water resistant and drought resistant, and has excellent growth. Most of it is orange-yellow or orange-red and contains rich nutrients such as vitamins, polysaccharides, proteins, amino acids, and trace elements [1,2]. Seabuckthorn is widely planted in northwest China. Seabuckthorn can survive on land with bad environments, so it is used for water and soil conservation. Seabuckthorn fruit itself is edible, but seabuckthorn fruit is not suitable for storage at room temperature. It usually appears on the market in the form of puree or dried fruit. In addition, seabuckthorn has high nutritional value, ecological value, and economic value [3].

Food processing is often a multi-stage process, and different processing methods need to be switched in different stages. Water content is an important parameter in food processing. Seabuckthorn fruits with different moisture contents are suitable for different processing processes and methods. Seabuckthorn fruits with different water content ranges will also affect their storage conditions. Therefore, the rapid, non-destructive, and real-time monitoring of the moisture/water content range of seabuckthorn fruit is very helpful for any further processing step. In this context, image detection often represents a fast, efficient, and low-cost method.

With the rapid development of the computer field in recent years, computer-related content has been widely used in the field of agricultural product processing. The concept of deep learning was put forward earlier, but there has been no major breakthrough in its application. With the improvement of computing power, applications began to develop rapidly. Deep learning can find the optimal structural parameters according to certain rules in training.

Traditional machine learning technology has limited ability in processing natural data in its original form. Deep learning allows computing models composed of multiple processing units to learn data representation with multiple abstract levels. Deep learning discovers complex structures in large datasets by using back-propagation algorithms to indicate how the machine should change its internal parameters [4].

Liang et al. [5] established a prediction model for moisture detection by collecting an image of the surface of tea and extracting the color component features of the image using nonlinear modeling methods. Chasiotis et al. [6] used ANN to model the moisture content of cylindrical quince slices in the drying process to predict the change in moisture content in the drying process.

Deep learning is a subfield of machine learning which attempts to learn high-level abstractions in data by utilizing hierarchical architectures [7]. In image target recognition and detection, its network model detection accuracy and generalization ability are improved compared with traditional machine learning, which is a new method. Deep learning has been widely used in the field of agricultural product detection [8], for example, the identification of plant diseases [9,10], crop variety categories [11], crop processing grade identification [12], quality detection [13,14,15], and adulterated meat detection [16].

Yang et al. [17] used hyperspectral imaging to detect the moisture content of carrots. Amjad et al. [18] used hyperspectral imaging to detect the moisture content of potato chips. The cost of water content detection by hyperspectral imaging is high, and the detection process and data analysis are relatively complex. It is difficult to realize real-time detection. Ramalingam et al. [19] collected images of wheat with different moisture contents using digital cameras and found that there was a linear relationship between wheat’s appearance and water content. It shows that there is a certain relationship between moisture content and material appearance. However, the standard of image acquisition is high.

Therefore, this study takes seabuckthorn fruit with different water contents as the research object and uses machine vision image detection technology to identify seabuckthorn fruit images with different water contents. The seabuckthorn fruit images with different water content ranges were modeled and trained by using transfer deep learning, aiming to provide a theoretical basis and new ideas for developing new water content detection technology.

## 2. Materials and Methods

### 2.1. Experimental Data

#### 2.1.1. Data Source (Experimental Materials)

A total of 100 kg of freshly frozen seabuckthorn fruits were purchased from the 170 Regiment of the 9th Division of the Corps in Tacheng, Xinjiang. The variety of seabuckthorn fruit belongs to “late autumn red”, which is planted locally in Xinjiang. It is usually harvested manually in the early morning of winter. After harvesting, it is quickly frozen in quick-freezing cold storage before noon and then stored in ordinary cold storage. It takes about 5 h from the cold storage in the production area to the school laboratory. Before the experiment, seabuckthorn fruit should be stored at minus 15 °C for 10 h. The seabuckthorn was preliminarily screened in the agricultural product-processing laboratory of the new engineering building in Beiyuan New District, Shihezi University, Xinjiang. The crushed and fractured seabuckthorn was separated, and the seabuckthorn with a good appearance was preserved in a refrigerator at minus 15 °C.

#### 2.1.2. Material Characteristics of Seabuckthorn

At a temperature of minus 15 °C, seabuckthorn fruit can perfectly preserve its appearance quality. Additionally, this temperature can protect seabuckthorn fruit from frost cracking. The basic physical parameters of normal seabuckthorn fruit materials are oval materials with an average long-axis grain size of about 12 ± 4 mm and a short-axis grain size of about 8 ± 2 mm. Seabuckthorn fruit contains vitamin C 210 ± 5 mg/g, total flavonoid content of 2.05 ± 0.04 mg RE/g DW, and total phenolic content of 1.40 ± 0.06 mg GAE/g DW. Seabuckthorn fruit is easy to be broken and oxidized when collected at room temperature. They can only be collected manually in winter and stored at low temperatures after quick-freezing in cold storage. They are easy to be oxidized and deteriorated when separated from branches.

#### 2.1.3. Experimental Instruments and Equipment

The drying equipment used in the experiment is an electric blast drying oven (DHG-9070A, Yiheng Technology Co., Ltd., Shanghai, China). It is used to heat the seabuckthorn and suck out the evaporated water. Maintain a temperature of 60 °C during drying. The wind is set to level 1. The device used for image acquisition is a 12-megapixel camera (aperture f/1.8, 12 million pixel camera, Sony, Beijing, China). This device is used for image acquisition. It collects images when weighing. Electronic balance (BSM-4220.4, Zhuojing Electronic Technology Co., Ltd., Shanghai, China) is used for weighing equipment. It is used to collect the total weight of seabuckthorn fruit and calculate the real-time moisture content. A layer of white steamer cloth (16 cm, Wuhu Silicone Products Co., Ltd., Wuhu, China) is paved on the dry material tray. Cut into the shape that conforms to the size of the material tray to carry seabuckthorn during the drying process. The white background is conducive to image acquisition and segmentation. All the above equipment is built on the experimental platform (Xinjiang Shihezi University) for further operation. The final data will be imported into a Hasaa portable computer (Hasaa Z7-KP7GZ, Shenzhen Shenzhou Computer Co., Ltd., Guangdong, China) for further processing.

#### 2.1.4. Image Acquisition

According to the requirements of seabuckthorn image acquisition, and in order to deal with various complex and changeable drying environments, a set of seabuckthorn fruit drying process image acquisition devices was designed. The experiment device consists of a drying oven (DHG-9070A), an electronic balance, a material tray, a silicone steamer mat, a computer, a light source, a silicone steamer mat, etc. (aperture f/1.8, 12 million pixel camera). The camera is located 50 cm above the material tray. The ring-led light source is used for lighting. The seabuckthorn fruit shall be defrosted and cleaned before drying, and the damaged and leaking seabuckthorn fruit will be screened out. Seabuckthorn fruit is placed above the tray, and a layer of a white fine mesh silicone steamer mat is laid at the bottom of the tray. During the drying process of the seabuckthorn fruit, images of seabuckthorn in the whole tray are collected. According to the water content gradient falling time measured in the pre-experiment, the weighing time interval is determined. When the set time interval is reached, the seabuckthorn fruit is taken out and weighed, and the image is collected on the image acquisition experimental platform. The water content calculation results correspond to the image collection until the water content drops to the target setting. The whole test process from the initial water content state to the target set water content is a round, with a total of six rounds. The number of seabuckthorn fruits in each round is about 120. In the process of image acquisition, the seabuckthorn fruit image at the edge is removed, and the seabuckthorn fruit image in the middle area is used as the experimental data.

#### 2.1.5. Division of Water Content Range

Tan et al. [20] studied the effects of different pretreatment methods on the quality of dried seabuckthorn fruit. The thawing drying method after pre-treatment will have a certain impact compared with the direct drying method. Therefore, seabuckthorn fruit after thawing was used as the research object in this study. The gradient of the water content range is defined according to the gradient of water content decline and the acquired database, which can avoid the problem of insufficient data due to preset gradient division. The final water content range is divided into six gradients, namely, 10–20%, 20–40%, 40–60%, 60–70%, 70–80%, and 80–90%, as shown in Figure 1.

#### 2.1.6. Training Environment Configuration

The operating system is Windows 10, the Matlab version is 2020 b, the operating environment is Intel i7-8750H, the basic frequency is 2.2 Ghz, the maximum frequency is 4.1 Ghz, and 16 GB of operating memory. The video card is the Geforce GTX1060 video card of NVIDIA, and the video memory version is 6 GB.

#### 2.1.7. Dataset Production

This model uses the collected dataset for training verification, in which 60% of the data are used as the training set, 20% of the data are used as the verification set, and 20% of the data are used as the test set.

A large amount of data will have a positive effect on deep learning. However, the advantage of transfer deep learning is that it can complete training through a smaller dataset. In the previous training process, it was found that if the number of each type was less than 80, the effect would not be ideal. The images of seabuckthorn fruit in each water content range are separated into a group of data, 150 in each group, with a total of 900 images.

The moisture content is calculated on a wet basis. The calculation equation is as follows.
(1)$$M=\frac{W}{W+D}\;$$
where *M* is the moisture content on a wet basis, *W* is the weight of water contained in the materials, and *D* is the weight of the dry matter contained in the material.

Record the acquisition time of each image and correspond the calculated water content with it.

In order to improve the training effect and stability, image data are generally enhanced during the training process. Hemanth et al. [21] proposed histogram equalization and other methods to improve image quality. Rahman et al. [22] adopted five image enhancement technologies to discuss the impact of image enhancement. This study also uses a part of image enhancement technology, mainly to expand the image set. In the training process, the dataset is expanded through Matlab’s data image enhancement processing, such as the random rotation of the X and Y axes, random rotation angle, random rescaling, the random horizontal translation of pixels, the random vertical translation of pixels, and other enhancement options. Image data enhancement can expand data to use current them efficiently [16].

### 2.2. Model Building

The model refers to the water content range detection model based on transfer deep learning. The image recognition technology based on deep learning does not need to manually extract specific features and only needs iterative learning to find appropriate features. This paper uses feature extractors from eight network models, including VGG-19, GoogLe-Net, Resnet-50, Resnet-101, Inception V3, Xception, MobileNet-v2, and Desnet-201, to build a detection network model, and conducts multiple training for each network model to achieve the best effect.

In the process of model building, the deep learning designer in Matlab was used. Considering the full embodiment of the actual seabuckthorn fruit characteristics, to achieve the full extraction of features, the seabuckthorn image was uniformly adjusted to a 300 × 300 × 3 RGB image. Under this size, the seabuckthorn fruit surface feature changes in the drying process can be displayed, so as to enhance the training effect.

Different models also have different training effects under the same training parameters, which is due to the impact of the network structure itself. We need to pay attention to the impact of the model structure at all times in the process of migration generalization.

#### 2.2.1. Network Composition Unit

The network models used in this paper are all based on the following units, and the detailed parameters of each unit in the network model are different. Due to the different connection modes, individual network models also need specific units to process the information transmitted from the upper layer. However, as a whole, some units are universal in the network and have the same functions.

ImageInputLayer: responsible for inputting images and performing normalization processing. [23,24].

ConvolutionLayer: the convolution layer performs the convolution operation on the image data. Slide the filter from the vertical and horizontal directions, calculate the weights, and add offset and output again. [25,26,27,28,29].

ReluLayer: the relu layer is to check element input and pooling [30]. This operation is equivalent to
$$f\left(x\right)=\{\begin{array}{c} x,x\geq 0\\ 0,x\leq 0 \end{array}$$

Clipped pooling is another pooling operation, which is used to prevent excessive data [31]. This operation is equivalent to:$$f\left(x\right)=\left\{\begin{array}{l} 0,x<0\\ x,0\leq x<ceiling\\ ceiling,x\geq ceiling \end{array}\right\}$$

DropoutLayer: The dropout layer will set some elements to 0 at random probability. [23,32].

BatchNormalizationLayer: Normalizing each input channel can speed up network training and reduce sensitivity [33].

CrossChannelNormalizationLayer: Implement cross-channel normalization [23].

PoolingLayer: The pooling layer divides the input into rectangular areas and calculates the specific value of each area to perform downsampling. These include average pooling, maximum pooling, and global average pooling [34].

AdditionLayer: summarize the inputs of multiple neural network layers.

DepthConcatenationLayer: accept inputs with the same height and width and connects along the channel dimension.

FullyConnectedLayer: the full connection layer is to multiply the input and weight matrix and then adds offset [28,29,35].

SoftmaxLayer: assign and output a probability value to each input [36].

ClassificationLayer: the cross entropy loss is usually calculated according to the softmax layer [36].

#### 2.2.2. Transfer Learning

Transfer learning is a machine learning method that transfers the characteristics of known training information from one application field to another [37]. It is a new machine learning method that uses existing knowledge to solve problems in different but related fields, which can cause the target domain to achieve better performance. In view of the strong performance and application of the deep learning model in other fields, this paper applies migration deep learning to the identification of the water content range of seabuckthorn fruit. In the field of computer vision, we usually use the parameters that have been trained in the Imagenet dataset [38] as the initial parameters of transfer learning to migrate to other fields for application.

The deep learning designer in Matlab is used to arrange the required network structure, adjust the model structure according to the required input and output, analyze the output network structure, adjust the problem structure after analysis, and then adjust the training parameters for multiple training.

The data determine the upper limit of the problem to be solved, whereas the model only approaches this upper limit. In other agricultural product image datasets, fine-tuning the network structure will also achieve different training effects [39].

#### 2.2.3. Network Structure Adjustment

Before the training of the model, due to problems such as the size of the network model structure, and the carrying capacity of the computer performance, it is necessary to adjust the graph size. Different network models require different input image sizes. Different models have specific requirements for the size of the input image. After the network structure is selected, the image usually needs to be processed to the required image size.

However, the respective image sizes from 224 × 224 × 3 of Vgg-19 to 299 × 299 × 3 of Xception have a certain impact on the full feature presentation of this study, and the distortion of image adjustment has a certain impact on the tiny feature presentation. Therefore, through the deep network designer of Matlab, the input layer images are uniformly adjusted to 300 × 300 × 3. Analyze the network model structure and adjust the conflicting layers to enable them to train the images required by this study. By comparing the influence of different input layer image sizes on the network accuracy, we can find the optimal training image size parameter to obtain the best training effect.

#### 2.2.4. Experimental Parameter Setting

The eight network models in this study all use the pre-training parameters of the thousand classification network models trained under the Image Net dataset, which can greatly save training time and computing resources. Due to the number of layers in the network structure and the different linking methods, the structure is different. The learning rate is trained in turn between 1×10^–6^–0.01, trying to find the best learning rate of each model for subtle features.

#### 2.2.5. Construction, Testing, and Application of Network Model

Based on the completion of the above preparations, the final adjustment of the network model structure is undertaken, mainly including the image input layer and the final classification layer [16]. The training process, testing process, and final application process of the network are shown in Figure 2.

## 3. Result and Discussion

### 3.1. Training Results

The higher the verification accuracy, the better the actual performance of the network. The training results show that the MobileNet-v2 network model has the best validation accuracy and test accuracy compared with the other seven models in this study, which indicates that the model has a better learning ability for seabuckthorn fruits with different moisture content.

#### 3.1.1. Model Accuracy and Training Parameters

Buiu C et al. [40] used MobileNet-V2 to realize the four classifications of cervical precancerous lesions, with an accuracy rate of 91.66%. Among the eight network models adopted in this paper (Table 1), the final verification accuracy can reach 95% after multiple adjustments and training. Among them, the VGG-19 and MobileNet-v2 network models have shown good results in multiple training, and the accuracy of multiple training can reach more than 95%. Among them, the VGG-19 network model is the best to reach 97.22%, and the MobileNet-v2 network model is the best to reach 98.61%. The latter is also the best effect among the eight network models adopted. It can be seen that the feature extraction layer of these two network models can comprehensively collect the subtle features of seabuckthorn image changes and show good results in the verification set.

#### 3.1.2. Visualization of Network Model Extraction Features

The convolutional neural network uses features to classify images, and the network learns these features by itself in the training process. In the learning process, we monitor the training progress and adjust the training parameters according to the features learned by the network layer. The features extracted [41] from each layer of the network model are related to the network structure and network unit parameters. During the training process, the network constantly adjusts its internal structural parameters to achieve the best training results. Due to a series of reasons such as the size of the convolution kernel and the number of convolution steps, the effect of feature extraction will be different. Because the appearance of seabuckthorn fruit is similar to an ellipse as a whole, the extracted features are mainly circular contours. Finally, we can see the image information of the seabuckthorn fruit water content range storage layer [16] through global synthesis.

### 3.2. Discussion on the Performance of Network Model

The network model has a certain general adaptability under the training of large datasets, but the visual image detection of seabuckthorn fruit moisture content requires that the feature extraction layer provided by the network model structure has a higher requirement for the perception of subtle structural changes. The change in similar water content ranges is not particularly significant in the appearance of seabuckthorn fruit, but the results show that the feature extraction layer of the above network model has the ability to identify subtle features.

When the number of model iterations is 300, the validation accuracy of DesNet-201 reaches 90% for the first time, but the final validation accuracy of Desnet-201 is 94.91%. The validation accuracy of the remaining seven models exceeded 95%. However, in general, GoogLe-Net converges the fastest, and the slowest model is VGG19, although VGG19 fluctuates more. MobileNet-v2 has steadily improved the verification accuracy during the training process and, finally, has obvious advantages in model size and training time.

The network size of the feature extraction layer based on the VGG-19 network model finally trained in this paper is about 730 M, whereas the network size of the feature extraction layer based on the MobileNet-v2 network model is about 13 M, and the network occupation volume is only 1.78%, which greatly reduces the memory occupation and computing resources occupied by program calls.

In addition, the average training time of 5 min based on the MobileNet-v2 network model is only 7.14% of the average training time of 70 min based on the VGG-19 network model. MobileNet-v2 has absolute advantages in memory occupation and training time. It also has certain advantages for further arrangement and application on embedded systems.

#### 3.2.1. The Influence of Network Model Training Fluctuation on Accuracy

Due to the correction of network unit parameters by backpropagation, the verification accuracy will fluctuate continuously during network training, as shown in Figure 3. From the perspective of verification accuracy, each network will have large fluctuations in the early training period and tend to be stable in the late training period. The fluctuation range in the later period is small, and the accuracy of the network will be relatively high. The trained network will perform better in the test. As the network tends to be stable and converges, it means that the feature extraction has been basically completed and good results have been obtained. Prolonging the training time may also provide better training results, but if the network has been violently fluctuating and does not converge, it means that the network cannot complete the training of such tasks.

#### 3.2.2. Influence of Different Network Model Structures on Network Model Performance

From the model based on the VGG-19 network feature extraction layer to the final selected model based on the MobileNet-v2 network feature extraction layer, the structure and depth of the network model are different, and the cross-layer connection mode is also different. From the 708-layer structure of the DesNet-201 network model to the 170-layer structure of MobileNet-v2, the effect of the image feature extraction of seabuckthorn fruit moisture change is different. Other network models used in this paper have hundreds of layers, but their performance in this dataset is still inferior to MobileNet-v2. There is no inevitable relationship between the accuracy of the network model and the depth of the network in the training process. It does not mean that the deeper the network, the better the effect. Different datasets need to try different network structures to find the most suitable one.

Therefore, in order to obtain ideal training results for different types of datasets, it is necessary to train a variety of different network structures to find the best network model. For specific datasets, the network structure can also be adjusted.

### 3.3. Model Validation

#### 3.3.1. Confusion Matrix

A confusion matrix is a standard format for precision evaluation [42]. It can be used to judge the performance of the network model. The confusion matrix of the final test of the eight network models used in this paper is shown in Figure 4. The MobileNet-v2 network model has the best test accuracy, which is 99.4%. Only one sample was predicted incorrectly, and the error occurred in the adjacent water content range.

The GoolgLe-Net network model has the fastest training speed, but the size of the training completion model is close to twice that of the MobileNet-v2 network model, and the lowest test accuracy is 91.7%. The VGG19 network model performs well, and its accuracy is second only to the MobileNet-v2 network model. However, the model occupies the most memory, and the training time is the longest. The ResNet-50, ResNet-101, Inception V3, and Xception network models perform well overall, but there are still differences between some predictions and actual values. The effect of the Desnet-201 network model in the verification process and testing process is not very good, and there are two data prediction values that are far from the actual value. Therefore, in general, the MobileNet-v2 network model performs best in the validation and testing process.

#### 3.3.2. Application of Water Content Range Detection

After the training test is completed, some images will be collected separately for program operation, and the operation result image is shown in Figure 5. Compared with the range of water content calculated in real time, the detection effect is ideal.

## 4. Conclusions

This paper studied the performance of eight network models based on fine-tuning deep learning on a seabuckthorn fruit dataset with subtle feature changes and finally concluded that the best network model for detecting the moisture content range of seabuckthorn fruit is fine-tuning MobileNet-v2. The MobileNet-v2 network model is superior to other network models in training speed and training accuracy, which shows that the model has an excellent feature extraction effect for subtle feature changes, and the accuracy of the verification set is 98.61%. It also performs well in the test process, and the accuracy rate of the test data is 99.4%. When using the trained network model, the average offline detection time is about 0.5 s, and the online detection time can be further shortened.

Using the method of transfer deep learning to detect the water content range of seabuckthorn fruit, the verification accuracy can reach 95% on eight network models, so the visual detection of seabuckthorn fruit water content ranges of transfer deep learning is a feasible method. In addition, the trained network can be directly used to realize the real-time detection of water content ranges of seabuckthorn fruit. The use of migration deep learning could also realize the moisture content range detection of other agricultural products, improve detection efficiency, and reduce detection costs.

## Figures and Tables

**Figure 1 foods-12-00550-f001:**
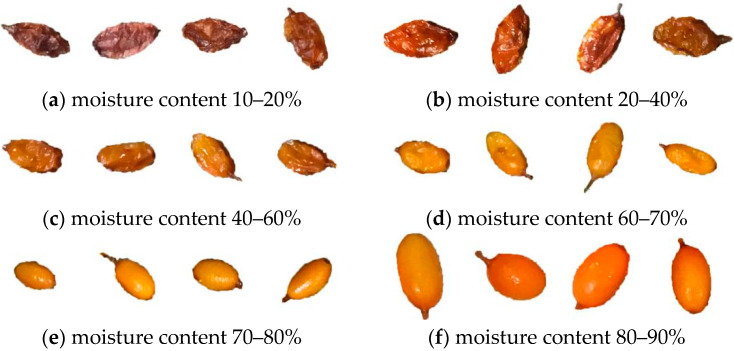
Six moisture content gradient images.

**Figure 2 foods-12-00550-f002:**
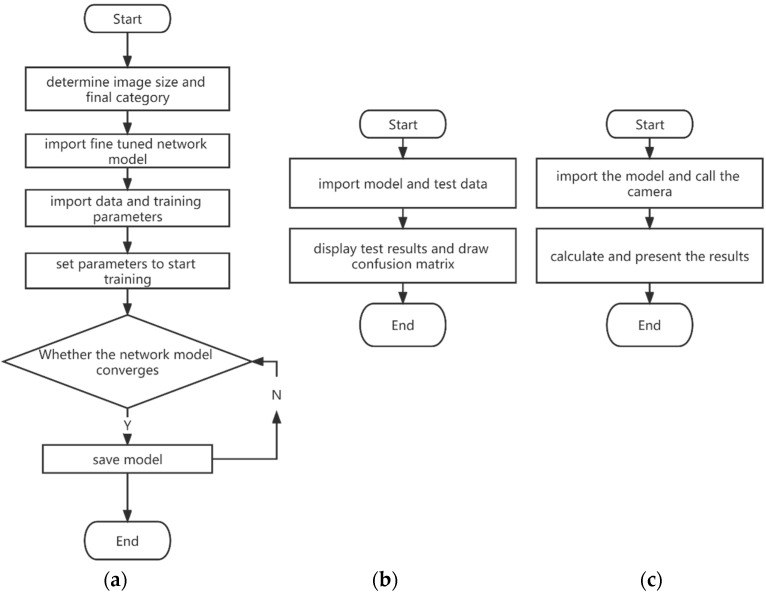
Process of model training, testing, and application. (**a**) Training process; (**b**) Test process; (**c**) Application process.

**Figure 3 foods-12-00550-f003:**
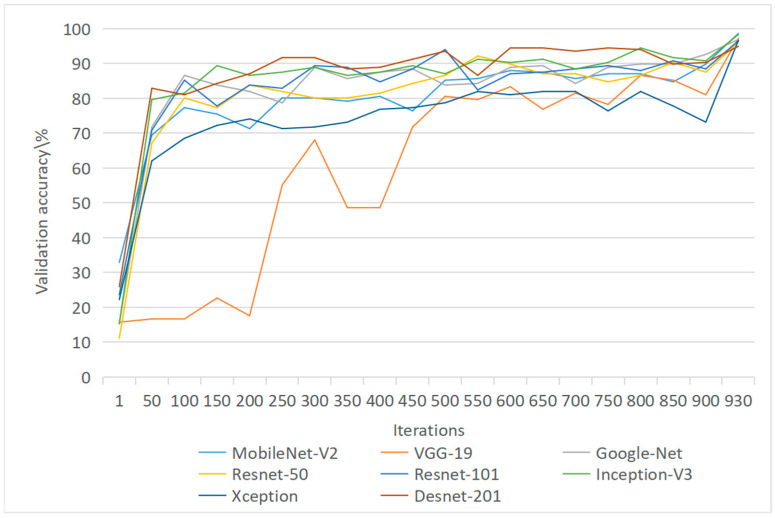
Network model training process.

**Figure 4 foods-12-00550-f004:**
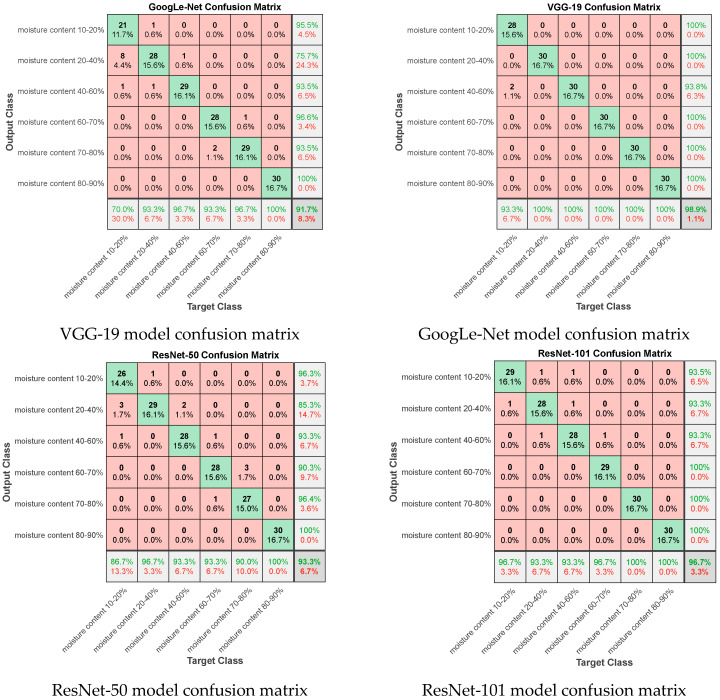
Test set confusion matrix.

**Figure 5 foods-12-00550-f005:**
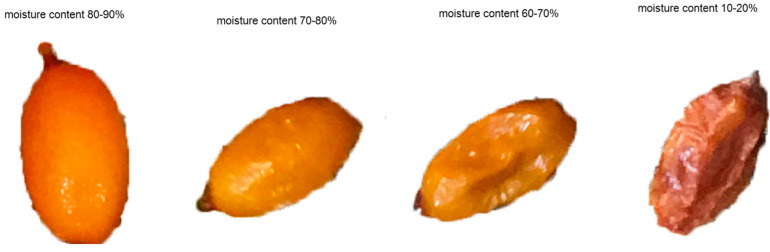
Display of program operation results.

**Table 1 foods-12-00550-t001:** Comparison results of models.

Network Model	Training Accuracy	Verification Accuracy	Average Model Size	Training Time of Optimal Accuracy Rate	Optimal Learning Rate
VGG19	100%	97.22%	731 M	138 min 43 s	0.0001
GoogLe-Net	99.68%	96.88%	24.9 M	4 min 57 s	0.0012
Resnet-50	98.75%	96.18%	91.0 M	14 min 1 s	0.001
Resnet-101	100%	96.53%	158 M	18 min 45 s	0.001
Inception-V3	99.89%	98.33%	84.9 M	16 min 56 s	0.002
Xception	98.87%	96.67%	81.4 M	20 min 21 s	0.004
MobileNet-v2	99.81%	98.69%	12.7 M	6 min 13 s	0.005
Desnet-201	97.35%	94.91%	73.4 M	80 min 38 s	0.005

## Data Availability

The datasets generated for this study are available upon request to the corresponding author.

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
