# Peer review of "Visual Detection of Water Content Range of Seabuckthorn Fruit Based on Transfer Deep Learning"

_foods, 2023, doi:10.3390/foods12030550_

Round 1
Reviewer 1 Report
The topic is potentially interesting considering the great improving that can originate from the use of rapid, non-invasive and automatic techniques to detect and quantify the water content range in fruits, as it is the method proposed in this study.
Without doubt, the topic will attract a great deal of industrial and academia interest on food quality control field, and a lot of work has been performed by the authors. However, I can not say that the manuscript is well written nor clearly structured and well organized.
The current version of the article has serious flaws in the way information is conveyed, structured and presented. There is a lack of details that, in my opinion, are important to help the reader in following the main logical thread of the work and to clearly understand all the work steps.
I strongly suggest authors considering the comments (listed below) raised during referee before resubmitting a new version of their manuscript.
GENERAL COMMENT
It seems that the manuscript has not been carefully reread before the submission: there are indeed some typos and grammatical errors. I strongly encourage authors to fix them during the manuscript revision.
Abstract
This first sentence seems to be incomplete (without verb). Please, verify and fix it.
Lines 21-22: "It" stands for "the model"? It should be better not starting sentences with "It". What do authors mean by "slight changes in characteristics"?
Line 22: authors here talk about "migration deep learning" while, before, they talk about just "models based on deep learning". Is it the same approach or is it another concept?
Introduction
Line 35: it is clear why, due to the explained characteristics, the plant of the Seabuckthorn fruit has high ecological value, while it is not so deductively explained its economic value: are these fruits edibles? Are they sold on large-scale retail trade or just in local, small market?
Line 36: this sentence seems to be a little bit "untied" with respect to the previous one. I mean, we were just talking about fruit/plant features and now we are already talking about data modelling... what in between? Why authors focused on using deep learning? Which data we are focused to model/predict? Why is it important to predict this/these parameter for the fruit?
I can see that the Introduction is currently too short since there is a serious lack of information before talking about the data modelling approach. Moreover, some other studies already attempt to do similar things with a similar approach? Or as well with different approach? I suggest authors to extend this section by providing the missing (but needed) information.
Materials and Methods
2.1.1: where do these samples were grown and harvested? All the samples belonged to the same cultivar/species? More information about the raw product should be added. How much time elapsed between sample's receipt/delivery and the analysis?
2.1.2: so, these fruits are normally quick frozen and then kept in cold storage: why the samples used in this study were instead kept frozen at -15 °C and not delivered just keeping them in cold storage?
2.1.3: it is clear that this section can not stand alone as it is currently presented: there is not even a verb to support the unique sentence written. Moreover, each one of the instrument and equipment listed deserves a little bit more of explanation on why was it used within the planned experimental flow (by the way, which one?) and which were the corresponding set parameters. Readers deserve to know deep details about what was it done, in order to be able to reproduce the same experiments, if needed. Without doubts, this could not be possible based on the actual information provided. I see that some more details are given in the following section, but it is still confusing. Thus, I strongly suggest authors to "fuse" section 2.1.3 and 2.1.4. by providing a clear description of the experimental flow (what was it done in sequence) and a careful description of each one of the process steps, together with an exhaustive explanation of the instrument’s parameters and settings.
2.1.4: samples have been received and stored at freezing temperatures: the drying process was thus started directly on the frozen samples or were them kept at room temperature for a brief period before starting the drying process? Please, clarify this aspect. Moreover, maybe that a schematic representation of the instrument setup described in this section could help the reader to better understand it.
2.1.5: was it this "division of water content range" established and achieved by the authors in their experiments or is it defined based on commercial parameters? Moreover, how authors considered the comparability between the drying process conducted on a fresh fruit and on a frozen one? It seems to me that some differences in the chemical-physical composition of the same fruit can potentially occur between fresh and frozen samples. Have authors considered these aspects? Are there some evidence that could be explained, or some studies that already considered this important aspect?
2.1.7: again, more details should be provided: why was the dataset expanded? Why 900 images? Where does this number come from? Which were the "already-known" differences about the imaged fruits? How the reference values of the humidity, or humidity range, were determined and collected? Provide references for the data image enhancement processing methods listed in the last sentence.
2.2.1: at the opposite, this section seems to be overly detailed. Authors should consider shrinking it by just reporting the essential information.
Result and Discussion
GENERAL COMMENT: my impression is that authors often alternate results with discussions, even into a same section. I suggest authors to first clearly and thoroughly present the results obtained, and then discuss them by reporting, when needed, references of similar works, by also discussing the comparison between these works and the present study.
Authors show Fig. 3 and Fig. 4 as a kind of result's maps, but these can not be understood just based on the information currently provided both in the text and in the captions.
3.3: why just the prediction results achieved on the final test data were presented? What about the results achieved on the training set and the "verification" (better validation) set? Confusion matrices should be replaced by a Table that summarize the values of the parameters considered to assess the model's classification performance (Accuracy, Sensitivity, Specificity ...?) for each modelling step i.e., training, first validation and final validation step. The confusion matrices could be presented as well, but separately (I suggest to provide them as Supplementary). Moreover, the performances of each model and their comparison are not enough discussed.
Line 319: the expression "best test effect" is not so deductive: better replacing it with the name of the performance indicator i.e., "accuracy".
Conclusions
Line 347: which models? Of those presented in Fig. 6, only 5 up 8 achieved over 95% of accuracy, not all 8. Please, clarify.
Line 351: "...can elso realize" should be replaced by "...could also realize".
Reviewer 2 Report
- The necessity of the visual detection of water content range of seabuckthorn fruit has not been clarified.
- An explanation of similar visual detection methods of water content of other fruits should have been reviewed.
- Similar deep learning procedures used to classify fruit products could have been reviewed.
- The deep learning method should have been compared to other more simple statistical traditional classification methods (as PCA) to demonstrate de advantage of deep learning.
- The further purpose of this classification method should be explained (food industry use).
- Lines 20-22: “It has a good detection effect for seabuckthorn fruits with different moisture content ranges with slight changes in characteristics”. Other physical/chemical characteristics of the fruits should have been measured apart from moisture content (diameters, texture parameters, sugar content,…).
- Line 60-62: the number of fruits should be added (in the abstract it is said “900 images…”, but the number of fruits is not clear).
- Lines 66-69: The basic physical parameters of the fruits (average and deviation values should be added (size, texture, sugar content,…) in order to validate this sentence.
- Lines 69-70: “Seabuckthorn fruit is easy to be broken and oxidized when collected at room temperature”. How was the state (absence of damages) of the fruit confirmed?
- Lines 70-71: “They can only be collected manually in winter and stored at low temperature 70 after quick freezing in cold storage”. The harvesting and storage conditions of the fruits used in the test should be defined.
- Line 71-72: “They are easy to be oxidized and deteriorated when separated from branches“. Why were the fruit frozen? Did it not affect the texture of the product and the dehydration process? The time taken from harvesting to freezing and to testing should be explained.
- Another set (or sets) of fruits from different origin could have been added to the model (specially to the validation set).
- Figure 6: it seems that a simpler statistical test using L a b color coordinates, size and shape could be used to segregate the different moisture level. A comparison to other analysis method to show the advantage of the proposed deep learning method should be added.
- Figure 6: clear differences in the percentages of the different moisture level categories are shown in the confusion matrix for the different network methods. It is crucial to explain these differences and discuss the advantages and disadvantages of the low percentages in certain moisture level categories, and define the most suitable network method. It is necessary to define in which moisture category is crucial to have a low classification error (a high percentage) due to the necessity of segregate the moisture level from a practical purpose.
- The conclusions should provide the practical usefulness off the proposed method and focus on the advantage of deep learning compared to other simpler methods.
Round 2
Reviewer 1 Report
I appreciate the revising work performed by the authors. I thank authors to take into account each section of the listed comments.
Something is already not clear and/or has to be further completed. Please refer to the feedback/response below.
Sincerely yours,
Thank you
---------------------------------------------------------
Dear Reviewer:
Thank you for your comments concerning our manuscript entitled “Visual detection of water content range of seabuckthorn fruit based on transfer deep learning” (No.: foods-2106511.). Those comments are all valuable and very helpful for revising and improving our paper. We have studied comments carefully and have made correction which we hope meet with approval. The main corrections in the paper and the response are as following:
Abstract
Point 1: This first sentence seems to be incomplete (without verb). Please, verify and fix it.
Response:
Thanks for your careful checks. We have adjusted the first sentence. The modified statement is as follows:
To realize the classification of sea buckthorn fruits with different water content ranges, a convolution neural network(CNN) detection model of sea buckthorn fruit water content range was constructed.
Rewiewer’s response: now it is OK. Thank you.
Point 2: Lines 21-22: "It" stands for "the model"? It should be better not starting sentences with "It".
What do authors mean by "slight changes in characteristics"?
Response:
Thanks for your careful checks.Yes, "It" stands for “the model”. We have checked and replaced it with “the model”.
“Slight changes in characteristics” means that the visual changes of seabuckthorn fruits in the adjacent moisture content range are not particularly obvious.
Rewiewer’s response: OK. But it would be better adding this explanation (obviously, in a compressed form) also in the text.
Point 3: Line 22: authors here talk about "migration deep learning" while, before, they talk about just "models based on deep learning". Is it the same approach or is it another concept?
Response:
We thank you for pointing this out. The same approach, but with some improvements.
We think that migration deep learning can be seen as a branch of models based on deep learning. It can solve the problem of small data sets, similar to transfer learning. The methods used are essentially the same. However, the training with pre training parameters can greatly improve the accuracy and reduce the training cost.
Rewiewer’s response: now it is OK. Thank you.
Point 4: Line 35: it is clear why, due to the explained characteristics, the plant of the Seabuckthorn fruit has high ecological value, while it is not so deductively explained its economic value: are these fruits edibles? Are they sold on large-scale retail trade or just in local, small market?
Response:
Thank you for your helpful suggestion. Seabuckthorn fruit is an edible berry with high nutritional value and they are usually sold in the local market.
Rewiewer’s response: now it is OK. Thank you.
Point 5: Line 36: this sentence seems to be a little bit "untied" with respect to the previous one. I mean, we were just talking about fruit/plant features and now we are already talking about data modelling... what in between? Why authors focused on using deep learning? Which data we are focused to model/predict? Why is it important to predict this/these parameter for the fruit?
Response:
We thank you for your valuable suggestion. We have supplemented some relevant contents and explained the importance of predicting the moisture content of materials. Below is the modified statement:
The seabuckthorn fruit images with different water content ranges were modeled and trained by using transfer deep learning. Seabuckthorn fruits with different moisture content are suitable for different processing processes and methods. Seabuckthorn fruits with different water content range will also affect the storage conditions. Therefore, rapid non-destructive detection of the moisture content range of seabuckthorn fruit is very helpful for further processing.
Rewiewer’s response: good about the added information, but again not ok the order by which the added information is presented. First, authors must talk about the reasons behind using image analysis followed by image elaboration. Secondly, they will talk about techniques already exploited and previous studies performed by different authors. Below, I propose a kind of suggestion based on the sentences already added by authors. After line 36, the text should be rearranged as follow:
“Food processing is often a multi-stage process, and different processing methods need to be switched in different stages. Water content is an important parameter in food processing. Seabuckthorn fruits with different moisture content are suitable for different processing processes and methods. Sea buckthorn fruits with different water content range will also affect the storage conditions. Therefore, rapid, non-destructive and real-time monitoring of the moisture/water content range of seabuckthorn fruit is very helpful for any further processing step. In this context, image detection often represent a fast, efficient and low-cost method.”
Ok then using text inserted from line 37 until line 42. Subsequently, from line 49 to line 73.
Then, the aim of the study is presented: “Therefore, this study takes seabuckthorn fruit with different water content as the research object, and uses machine vision image detection technology to identify seabuckthorn fruit images with different water content. The seabuckthorn fruit images with different water content ranges were modeled and trained by using transfer deep learning, aiming to provide theoretical basis and 80 new ideas for developing new water content detection technology.”
Point 6: I can see that the Introduction is currently too short since there is a serious lack of information before talking about the data modelling approach. Moreover, some other studies already attempt to do similar things with a similar approach? Or as well with different approach? I suggest authors to extend this section by providing the missing (but needed) information.
Response:
Thank you for your helpful advice. We have added some relevant content and described the necessity of this study, as shown below:
Yang et al.[38] used hyperspectral imaging to detect the moisture content of carrots. Amjad et al.[40] used hyperspectral imaging to detect the moisture content of potato chips. The cost of water content detection by hyperspectral imaging is high, and the detection process and data analysis are relatively complex. It is difficult to realize real-time detection. Ramalingam et al. [39]collected images of wheat with different moisture content using digital cameras, and found that there was a linear relationship between wheat appearance and water content. It shows that there is a certain relationship between moisture content and material appearance. But the standard of image acquisition is high.
Food processing is often a multi-stage process, and different processing methods need to be switched in different stages. Water content is an important parameter in food processing. It is of great significance to realize real-time monitoring of water content range in the production process. Image detection is often a fast, efficient and low-cost method.
Rewiewer’s response: now it is OK. Thank you.
Point 7: 2.1.1: where do these samples were grown and harvested? All the samples belonged to the same cultivar/species? More information about the raw product should be added. How much time elapsed between sample's receipt/delivery and the analysis?
Response:
Thank you for your helpful advice. We have made the following supplements, as shown below:
A total of 100kg fresh frozen seabuckthorn fruits were purchased from the 170 Regiment of the 9th Division of the Corps in Tacheng, Xinjiang. The variety of seabuckthorn fruit belongs to "late autumn red", which is planted locally in Xinjiang. It is usually harvested manually in the early morning of winter. After harvest, it is quickly frozen in the quick freezing cold storage before noon, and then stored in ordinary cold storage. It takes about 5 hours from the cold storage in the production area to the school laboratory.
Rewiewer’s response: now it is OK. Thank you.
Point 8: 2.1.2: so, these fruits are normally quick frozen and then kept in cold storage: why the samples used in this study were instead kept frozen at -15 °C and not delivered just keeping them in cold storage?
Response:
We thank you for pointing this out. We have added some relevant content, as shown below:
At a temperature of minus 15 ℃, seabuckthorn fruit can perfectly preserve its appearance quality. And this temperature can protect seabuckthorn fruit from frost cracking. In addition, insufficient freezing temperature may cause adhesion.
Rewiewer’s response: now it is OK. Thank you.
Point 9: 2.1.3: it is clear that this section can not stand alone as it is currently presented: there is not even a verb to support the unique sentence written. Moreover, each one of the instrument and equipment listed deserves a little bit more of explanation on why was it used within the planned experimental flow (by the way, which one?) and which were the corresponding set parameters. Readers deserve to know deep details about what was it done, in order to be able to reproduce the same experiments, if needed. Without doubts, this could not be possible based on the actual information provided. I see that some more details are given in the following section, but it is still confusing. Thus, I strongly suggest authors to "fuse" section 2.1.3 and 2.1.4. by providing a clear description of the experimental flow (what was it done in sequence) and a careful description of each one of the process steps, together with an exhaustive explanation of the instrument’s parameters and settings.
Response:
Thank you for your helpful suggestion and careful checks. We have added explanations on the use and settings of each experimental equipment. The modification results are as follows:
DHG-9070A electric blast drying oven: It is used to heat seabuckthorn and suck out the evaporated water. Maintain a temperature of 60 ° C during drying. The wind is set to level 1.
12 megapixel camera: This device is used for image acquisition. It collects images when weighing.
Electronic balance: It is used to collect the total weight of seabuckthorn fruit and cal-culate the real-time moisture content.
White steamer cloth: Cut into the shape that conforms to the size of the material tray to carry seabuckthorn during the drying process. The white background is conducive to image acquisition and segmentation.
Experimental platform (Xinjiang Shihezi University): The experimental platform is used to place various experimental equipment.
Hasaa portable computer: The computer is used for image acquisition and further depth processing.
Rewiewer’s response: OK. But I think it would be better re-organize this part as a sequence of sentences and not as a kind of bullet-pointed list. Some more specifications could be reported yet for each equipment mentioned.
Point 10: 2.1.4: samples have been received and stored at freezing temperatures: the drying process was thus started directly on the frozen samples or were them kept at room temperature for a brief period before starting the drying process? Please, clarify this aspect. Moreover, maybe that a schematic representation of the instrument setup described in this section could help the reader to better understand it.
Response:
We thank you for pointing this out. The seabuckthorn fruit shall be defrosted and cleaned before drying, and the damaged and leaking seabuckthorn fruit will be screened out.
The image acquisition device is mainly composed of a camera and an annular light source, and the others are auxiliary equipment. The camera collects images 50cm above the material.
Rewiewer’s response: now it is OK. Thank you.
Point 11: 2.1.5: was it this "division of water content range" established and achieved by the authors in their experiments or is it defined based on commercial parameters? Moreover, how authors considered the comparability between the drying process conducted on a fresh fruit and on a frozen one? It seems to me that some differences in the chemical-physical composition of the same fruit can potentially occur between fresh and frozen samples. Have authors considered these aspects? Are there some evidence that could be explained, or some studies that already considered this important aspect?
Response:
Thank you for pointing this out. "division of water content range" is based on the uniform division of the gradient of the range of decreasing water content during the experiment. This can ensure the uniformity of data. Seabuckthorn fruit can be stored at room temperature for a long time when it is dried to 10-20% moisture content. This range can be used as the drying end point, in which case the drying process is ended.
It can be seen from “section 2.1.4” that sea buckthorn fruit can be fully thawed before drying. In addition, the frozen seabuckthorn fruit cannot be confirmed to be complete and undamaged.
Rewiewer’s response: OK, but still incomplete. Authors do not give evidence that the followed procedure is actually the “standard” procedure. Did some other work deal with the drying of fresh or frozen/thawed sea buckthorn fruit? Please, clarify this aspect. If so, appropriate references should be added.
Point 12: 2.1.7: again, more details should be provided: why was the dataset expanded?
Why 900 images?
Where does this number come from?
Which were the "already-known" differences about the imaged fruits?
How the reference values of the humidity, or humidity range, were determined and collected? Provide references for the data image enhancement processing methods listed in the last sentence.
Response:
Thank you for your valuable suggestion. Expanding the data set can improve the training effect and stability. In the previous training process, it was found that if the number of each type was less than 80, the effect would not be ideal. In the training process of this study, 150 images were prepared for each category, so there were a total of 900 images. The surface folds and colors of seabuckthorn fruits with different water content are different.
Rewiewer’s response: OK only for the first 2 points. Information about how the reference values of humidity range were determined, and references for the data image enhancement processing methods are still missing.
Point 13: 2.2.1: at the opposite, this section seems to be overly detailed. Authors should consider shrinking it by just reporting the essential information.
Response:
Thank you for your helpful advice. We have simplified the relevant content in our manuscript.
Rewiewer’s response: now it is OK. Thank you.
Point 14: GENERAL COMMENT: my impression is that authors often alternate results with discussions, even into a same section. I suggest authors to first clearly and thoroughly present the results obtained, and then discuss them by reporting, when needed, references of similar works, by also discussing the comparison between these works and the present study.
Response:
Thank you for your valuable suggestion. After section “3.1”, we have made a summary. Below is the modified statement:
The higher the verification accuracy, the better the actual performance of the network. The training results show that the MobileNet-v2 network model has the best validation accuracy and test accuracy compared with the other seven models in this study, which indicates that the model has better learning ability for seabuckthorn fruits with different moisture content.
Rewiewer’s response: OK.
Point 15: Authors show Fig. 3 and Fig. 4 as a kind of result's maps, but these can not be understood just based on the information currently provided both in the text and in the captions.
Response:
We thank you for pointing this out. We have added explanations for Fig. 3 and Fig. 4.
Fig. 3 mainly shows some features extracted from the network. Because of the network settings, the extracted features will be different. Therefore, only some features extracted from the training process of the best network are displayed. Fig. 4 shows the final synthetic texture information, and its differentiation can prove that the model has learned the differences between various types. Below is the modified statement:
During the training process, the network constantly adjusts its internal structural parameters to achieve the best training results. Due to a series of reasons such as the size of convolution kernel and the number of convolution steps, the effect of feature extraction will be different. Therefore, the best network model is selected to show some features extracted from the model. As shown in Figure 3. Because the appearance of seabuckthorn fruit is similar to an ellipse as a whole, the extracted features are mainly circular contours. And finally, we can see the image information of seabuckthorn fruit water content range storage layer [33] through global synthe-sis, as shown in Figure 4. The six images cover the summary of texture features of sea-buckthorn fruits in six moisture content ranges. Because of the repeated overlapping of textures, it may be difficult to distinguish the final image. However, the final image can show that there is an obvious texture gap between different types, which proves that the network has learned various features to detect different moisture content.
Rewiewer’s response: OK. But I still think that these 2 images are quite complex to be understood by the reader. Therefore, they could be removed or, at least, to be given as Supplementar material.
Point 16: 3.3: why just the prediction results achieved on the final test data were presented?
What about the results achieved on the training set and the "verification" (better validation) set?
Response:
We thank you for pointing this out. The accuracy rate in Table 2 and Figure 5 is the verification accuracy rate. The training accuracy rate is only a reference for the training process, so it is not included in the article. The final performance of the network model depends on the accuracy of the test data.
Rewiewer’s response: I agree that the results obtained on the test data definetely prove the model performance and accuracy. But it is also true that a comparison with the training set’s statistics is commonly given as comparison with the validation’s ones, and in order to prove the robustness and reliability of the performed models.
Point 17: Confusion matrices should be replaced by a Table that summarize the values of the parameters considered to assess the model's classification performance (Accuracy, Sensitivity, Specificity ...?) for each modelling step i.e., training, first validation and final validation step. The confusion matrices could be presented as well, but separately (I suggest to provide them as Supplementary). Moreover, the performances of each model and their comparison are not enough discussed.
Response:
We thank you for your valuable suggestion. We further analyzed the confusion matrix and added the following contents:
The GoolgLe-net network model has the fastest training speed, but the size of the training completion model is close to twice that of the MobileNet-v2 network model, and the lowest test accuracy is 91.7%. The VGG19 network model performs well, and its accu-racy is second only to MobileNet-v2 network model. However, the model occupies the most and the training time is the longest. ResNet-50, ResNet-101, Inception V3, and Xcep-tion network models perform well overall, but there are still differences between some predictions and actual values. The effect of Desnet-201 network model in the verification process and testing process is not very good, and there are two data prediction values that are far from the actual value. Therefore, in general, the MobileNet-v2 network model per-forms best in the validation and testing process.
Rewiewer’s response: OK.
Point 18: Line 319: the expression "best test effect" is not so deductive: better replacing it with the name of the performance indicator i.e., "accuracy".
Response:
Thank you for your helpful advice. We have replaced it with “best test accuracy”.
Rewiewer’s response: now it is OK. Thank you.
Point 19: Line 347: which models? Of those presented in Fig. 6, only 5 up 8 achieved over 95% of accuracy, not all 8. Please, clarify.
Response:
We thank you for pointing this out. There are some problems in expression. Figure 6 shows the data on the final test set. This can represent the actual performance of the network model in the final application.
The eight network models can reach 95% or more in the end, which means the verification accuracy. In the conclusion, the accuracy rate is modified to "verification accuracy ".
Rewiewer’s response: still not clear. Moreover, please be more specific with caption of Figure 6: confusion matrices obtained on the “training ”/ “test” / “verification” samples set.
Point 20: Line 351: "...can elso realize" should be replaced by "...could also realize".
Response:
Thanks for your careful checks. We have replaced "...can elso realize " with "...could also realize ".
Rewiewer’s response: now it is OK. Thank you.
Author Response
请参阅附件。

Reviewer 2 Report
The authors have clearly responded to the reviewer´s comments.
There are still some questions that could be addressed:
The further purpose of the proposed classification method specifically for seabuckthorn fruit should be explained (food industry use).
The number of tested fruits and rounds could be clarified and remarked. 720 seabuckthorn fruits were tested in 6 rounds, 120 fruits per round and 150 images per round. How many images were taken per fruit?
Besides fruit size, other basic parameters from the fruits could be added (texture, sugar content…).
It is still not clear how the absence of damages of the fruit was confirmed. Was it visually tested?
The storage conditions of the fruits used in the test have been explained. However, storage times and temperatures should be added.
It has been explained that in the six rounds, "late autumn red" variety was mainly used, but some other varieties were mixed. The proportion of the different varieties should be added, specially to undestand the different proportions in the model and the validation sets.
Author Response
Dear Reviewer:
Thank you for your comments concerning our manuscript entitled “Visual detection of water content range of seabuckthorn fruit based on transfer deep learning” (No.: foods-2106511.). Those comments are all valuable and very helpful for revising and improving our paper. We have studied comments carefully and have made correction which we hope meet with approval. The main corrections in the paper and the response are as following:
Point: The further purpose of the proposed classification method specifically for seabuckthorn fruit should be explained (food industry use).
Response:
We thank you for pointing this out. We have made adjustments to the introduction, in which we have further explained the use of classification.
Food processing is often a multi-stage process, and different processing methods need to be switched in different stages. Water content is an important parameter in food processing. Seabuckthorn fruits with different moisture content are suitable for different processing processes and methods. Seabuckthorn fruits with different water content range will also affect the storage conditions. Therefore, rapid, non-destructive and real-time monitoring of the moisture/water content range of seabuckthorn fruit is very helpful for any further processing step. In this context, image detection often represent a fast, efficient and low-cost method.
Point: The number of tested fruits and rounds could be clarified and remarked. 720 seabuckthorn fruits were tested in 6 rounds, 120 fruits per round and 150 images per round. How many images were taken per fruit?
Response:
The images of each round are collected as a whole. The image of each seabuckthorn fruit is separated separately in the later stage. During the whole drying process, about 20 images were collected for each fruit.
Point: Besides fruit size, other basic parameters from the fruits could be added (texture, sugar content…).
Response:
We thank you for your valuable suggestion. We have added the relevant content in our manuscript. Below is the modified statement:
Seabuckthorn fruit contains vitamin C 210 ± 5mg/g, the total flavonoids content of 2.05 ± 0.04 mg RE/g DW and total phenolic content 1.40 ± 0.06 mg GAE/g DW.
Point: It is still not clear how the absence of damages of the fruit was confirmed. Was it visually tested?
Response:
Thank you for your helpful suggestion.Yes, we conducted a visual inspection.We will select seabuckthorn fruit before drying. Damaged seabuckthorn juice will flow out, and we will pick it out.
Point: The storage conditions of the fruits used in the test have been explained. However, storage times and temperatures should be added.
Response:
Thank you for your helpful suggestion. We have added the relevant content in our manuscript. Below is the modified statement:
Before the experiment, seabuckthorn fruit should be stored at minus 15 ℃ for 10 hours.
Point: It has been explained that in the six rounds, "late autumn red" variety was mainly used, but some other varieties were mixed. The proportion of the different varieties should be added, specially to undestand the different proportions in the model and the validation sets.
Response:
We thank you for pointing this out. The proportion of other varieties is very low. Several hundred seabuckthorn fruits may contain several, and their appearance characteristics are very similar. The final impact on the model is very small.